

# Observational evidence of particle condensational growth in the UTLS over Tibetan Plateau

**Qianshan He[1,2], Jianzhong Ma[3], Xiangdong Zheng[3], Xiaolu Yan[3], Holger Vömel[4], Frank G. Wienhold[5], Wei Gao[1,2], Dongwei Liu[1,2], Guangming Shi[6], Tiantao Cheng[7]**

[1]Shanghai Meteorological Service, Shanghai, China
[2]Shanghai Key Laboratory of Meteorology and Health, Shanghai, China
[3]State Key Laboratory of Severe Weather & CMA Key Laboratory of Atmospheric Chemistry, Chinese Academy of Meteorological Sciences, Beijing, China
[4]Earth Observing Laboratory, National Center for Atmospheric Research, Boulder, CO, USA
[5]ETH Zurich,Institute for Atmospheric and Climate Science (IAC),CH-8092 Zurich,Switzerland.
[6]Chongqing Institute of Green and Intelligent Technology, Chinese Academy of Sciences, Chongqing, China
[7]Department of Atmospheric and Oceanic Sciences, Institute of Atmospheric Sciences, Fudan University, Shanghai, China

Correspondence to: Jianzhong Ma (majz@cma.gov.cn)

**Key Points:**

1. Balloon-borne measurements show an enhanced aerosol layer consisting dominantly of fine particles in the UTLS over Tibetan Plateau.

2.Water vapor plays an important role in the growth of these fine particles.

**Abstract**

We measured the vertical profiles of aerosol backscattering ratio (BSR) with a balloon-borne lightweight COBALD at Linzhi, located in the southeastern Tibetan Plateau, in the summer of 2014. An enhanced aerosol layer in the upper troposphere/lower stratosphere (UTLS), with BSR (455 nm)>1.1 and BSR (940 nm)>1.4, was observed. The Color Index (CI) of the enhanced aerosol layer, defined as the ratio of aerosol backscatter ratios at wavelengths of 940 nm and 455 nm, varied



from 4 to 8, indicating the prevalence of dominant fine particles with mode radius less than 0.1 μm. We find that except for the very small particles (mode radius smaller than 0.04 μm) at low relative humidity (RHi < 40%), the relatively large particles in the aerosol layer were generally very hydrophilic as their size increased dramatically with relative humidity. This result indicates that water vapor can play a very important role in the formation of large amounts of fine particles in the UTLS over the Tibetan Plateau. Our observations provide observation-based evidence supporting that the aerosol particle condensational growth is an important process for the summer ATAL enhancement over the Tibetan Plateau.

**Keywords:** ATAL, condensational growth, COBALD

## 1. Introduction

The Asian Tropopause Aerosol Layer (ATAL) extends over a large area within the Asian summer monsoon circulation and may significantly influence ozone, cirrus clouds and global climate by chemical, micro-physical and radiative processes [Gettelman et al., 2011; Vernier et al., 2011; Fadnavis et al., 2013; Thomason and Vernier, 2013; Vernier et al., 2015]. Particles in the ATAL are likely to be lifted to the lower stratosphere by the large-scale upward circulation within the south Asian anticyclone, and then influence the aerosol amount in the global stratosphere significantly [Park et al., 2007]. Solomon et al. [2011] found that the radiative forcing of increased aerosols in the global stratosphere from 2000 to 2010 is -0.1W·m$^{-2}$, which weakened the global warm effect from greenhouse gases.

In addition to the maximum concentration of aerosols found in the ATAL as mentioned above, the concentrations of tropospheric trace gases (i.e., water vapor, CO, $CH_4$ and HCN) are higher within the Asian summer monsoon anticyclone than in surrounding regions, while the stratospheric trace gases (i.e, $O_3$, $HNO_3$ and HCl) are lower [Park et al., 2004; Randel et al., 2010]. Actually, the maximum aerosol concentration near the tropopause over the Tibetan Plateau has also been observed by lidar and balloon borne measurements [Kim et al., 2003; Tobo et al., 2007; He et al.,


2014]. Li [2005] showed that the aerosol plume is detectable in the anticyclone around the altitude of 150 hPa over the Tibetan Plateau through satellite observations and model study.

The formation mechanism of ATAL has not been fully understood mainly due to sparse in situ measurements. [Frey et al., 2011] proposed that nucleation events at very low temperatures accompanied by the outflow of convective systems could be dominant process in the production of ATAL. Vernier et al. [2015] found that there is a one-month phase lag of the aerosol scattering ratio from the Cloud-Aerosol Lidar and Infrared Pathfinder Satellite Observation (CALIPSO) after the relative humidity with respect to ice (RHi) from the Microwave Limb Sounding (MLS) at the beginning of the convective period (May/June), possibly due to the growth of the nanometric particles to the larger particles that can be detected by satellites.

Both condensation and coagulation contribute to the particle growth, even though these two processes are triggered by different mechanism. Model studies have shown that the coagulation is more important than the nucleation in the control of the number concentration of fine particles (with diameter larger than 10 nm) in the UTLS [English et al., 2011; Pierce and Adams, 2009; Timmreck et al., 2010]. Except for coagulation, the effect of condensation on particle growth is less documented in previous studies. Weigel et al. [2011] found that supersaturated gases, which can nucleate to form neutral and charged molecular clusters, also condense onto pre-existing aerosol particles and cloud droplets. Earlier studies demonstrated that the stratospheric aqueous $H_2SO_4$ aerosol can absorb a large amount of gaseous $HNO_3$ and $H_2O$ at temperatures (about 200K) between the nitric acid trihydrate (NAT) and ice frost points [Carslaw et al., 1994; Tabazadeh et al., 1994], leading to a steep increase in particle volume. Heterogeneous reactions are active on the extreme cold stratospheric aerosols and polar stratospheric clouds (PSCs) over the winter poles. These aerosols and PSCs are composed either of supercooled ternary solution (STS) droplets ($HNO_3 \cdot H_2O \cdot H_2SO_4$), ice particles or solid hydrates (most likely NAT) and can grow to larger particles that are easy to sediment [Voigt et al., 2008; Engel, 2013]. However, unlike the studies about PSCs, the growth mechanism of the particles in the ATAL is still poorly described





due to the lack of sufficient observations.

In-depth investigations on the aerosol size distribution, chemical composition and growth process are needed for a better understanding of the characteristics and formation mechanism of ATAL. It is difficult to obtain much more information merely by means of remote sensing measurements, such as satellite and lidar, because those sensors are not sensitive to ultra-fine particles. In such case, balloon and/or air borne *in situ* measurement provide an additional and even better tool for exploring the ATAL. Using a balloon-borne optical particle counter at Lhasa, China, Tobo et al. (2007) measured the vertical profiles of aerosols and found occurrences of relatively high number concentrations of sub-micron size aerosols near the tropopause region during the Asian summer monsoon period. They considered that the enhanced aerosol layer in the UTLS connected closely with the transportation of water vapor from the Asian summer monsoon. An increased amount of water vapor was found in the UTLS within the Asian summer monsoon anticyclone (Bian et al., 2012; Li et al., 2017). Recently, a climate model simulation demonstrated that the abundant anthropogenic aerosol precursor emissions from Asia coupled with rapid vertical transport associated with monsoon convection could lead to significant particle formation in the upper troposphere within the ASM anticyclone (Yu et al., 2017). More Recently, a series of balloon borne activities over India and Saudi Arabia during the Balloon Measurements of the Asian Tropopause Aerosol Layer (BATAL) campaigns revealed that the ATAL is composed of mostly small (r < 0.25 μm) liquid (~80%–95%) aerosols with the dominant composition of nitrate (Vernier et al., 2017).

As part of the project Tibetan Ozone, Aerosol and Radiation (TOAR) [see More Information on ACP Special Issue, available at: http://www.atmos-chem-phys.net/special_issue331.html], the vertical profiles of aerosols over the southeastern Tibetan Plateau were measured in June and July of 2014. In this paper, we present the results from balloon borne radiosonde measurements, and investigate the contribution of condensational growth by gas-to-particle conversion processes to the observed high concentrations of fine particles in the UTLS over the Tibetan Plateau.



## 2. Experiment

The field experiment was carried out at the Linzhi Meteorological Bureau (29.67°N, 94.33°E; 2992 m MSL), located in the southeastern Tibetan Plateau, from June 6 to July 31, 2014. During the field campaign, seven balloon sondes were launched, with each sounding taking place at about 16:00 UTC on June 18 (case 1), June 24 (case 2), July 6 (case 3), July 15 (case 4), July 21 (case 5), July 25 (case 6) and July 30 (case 7), respectively. The balloon sonde payload was composed of the Compact Optical Backscatter AerosoL Detector (COBALD) particle backscatter sonde, the IMet and RS92 radiosonde, and the cryogenic frost-point hygrometer (CFH). The payload was lifted by a 1600 g latex balloon, which flew at an ascent rate of 5-7 m s$^{-1}$. Data were obtained from the lunching point until an altitude between 30 km to 35 km where the balloon generally burst. In this study, only the ascending data are analyzed.

### 2.1 COBALD particle backscatter sonde

The lightweight COBALD, developed by Prof. Thomas Peter's group at ETH Zurich, uses two high power light emitting diodes (LEDs) operating at 455nm (blue) and 940nm (infrared) with a silicon detector averaging the light scattered back from molecules or aerosols at angles centered near 173° for typically one second time periods [Rosen and Kjome, 1991; Wienhold, 2012; Cirisan et al., 2014]. COBALD measurements are only carried out at local nighttime as daylight saturates the sensitive detector.

Backscatter ratios (BSR) at two wavelengths are retrieved from COBALD measurement, which is defined as follow,

$$BSR = \frac{\beta_a + \beta_m}{\beta_m} = \frac{N_a \cdot \sigma_a + N_m \cdot \sigma_m}{N_m \cdot \sigma_m} \qquad (1)$$

where $\beta$ denotes backscatter coefficient, $N$ the number concentration, and $\sigma$ the backscatter cross section. The subscripts $a$ and $m$ indicate contributions from aerosol particles and air molecules, respectively. The backscatter cross section for air molecules can be calculated from Rayleigh scatter theory and the number concentration for air molecules is derived from atmospheric pressure and temperature measured by the radiosonde. From the COBALD raw data the blue and infrared backscatter ratio of each



individual flight profile was derived with an accuracy of 5 % and the precision in an
order of 1% [Vernier et al., 2015]. The backscattering cross section for aerosol particles
can be calculated from Mie scatter theory for a specified effective radius. The aerosol
backscatter ratio (ABSR) is defined as,
$$ABSR = \frac{\beta_a}{\beta_m} = BSR - 1 \qquad (2)$$
The ABSR values at two wavelengths are used to calculate the Color Index [CI,
Rosen et al., 1997], which is defined as the ABSR at 940 nm divided by the ABSR at
455 nm. The CI is proportional to the ratio of the backscatter cross sections at 940 and
455 nm, and hence it can provide an estimate of the particle size. Assuming an index
of refraction of 1.45 with 75% sulfate and a typical lognormal size distribution of the
stratospheric aerosols [Rosen and Kjome, 1991], the backscatter cross sections $\sigma_a$ at the
wavelengths used by COBALD are calculated by Mie theory, and further the CI as a
function of the mean radius of total aerosol particles is derived. Because no information
on standard deviation of the lognormal distribution is available, the possible lower and
upper limits the standard deviation are assumed to be 1.8 and 2.2 [Deshler et al., 2003].
By comparing the observed CI with the calculated one for different standard deviations,
the range of possible mean radius can be obtained, and the number concentration and
further volume concentration for aerosol particles can be retrieved from the observed
ABSR according to the Equation (1).
**2.2 Radiosonde observations**
In this study we use the air temperature profiles from the RS92 radiosondes with
an uncertainty of $\pm0.2℃$ below 100 hPa and $\pm0.3℃$ between 100 and 20 hPa. The
profiles of water vapor are obtained from CFH measurements. The CFH is a
microprocessor-controlled instrument with a lightweight of 400 g, and it uses a
cryogenic liquid as cooling agent and operates based on the chilled-mirror principle
[Vömel et al., 2007a]. The uncertainty of frost point or dew point measured by the CFH
is smaller than 0.2 K. Correspondingly, the uncertainty in relative humidity is estimated
to be 2 % for measurement in the lower troposphere and 5 % in the tropical tropopause
region [Vömel et al., 2016]. As a standard for water vapor measurements, CFH has





been used in numerous intercomparison experiments, such as the validation of Aura
Microwave Limb Sounder (MLS) water vapor products, globally [Vömel et al., 2007b]
and specifically over the Tibetan Plateau [Yan et al., 2016].
**3. Results and discussion**

6       Figure 1 shows the BSR profiles at two wavelengths and calculated CI profiles

from COBALD measurement, as well as the profiles of temperature and RH over ice
respectively from RS92 and CFH measurement for three typical cases on June 18, July
15 and 25, 2014. The COBALD measurements suggest an enhanced aerosol layer (BSR
(455 nm)>1.1 and BSR (940 nm)>1.4) extending from 200 hPa (~12 km) to 10hPa (~28
km) with a maximum above the tropopause (90 hPa, ~17 km). The RHi near the
maximum of the enhanced aerosol layer varies from 30% to 40%, indicating that it is
impossibly caused by cirrus cloud, which cannot persist at these dry conditions. It
should be noted that the volcanic eruption might have negligible influence on the
observed aerosol layers as no volcanic eruption occurred during the experiment period.
The calculated CI of the enhanced aerosol layer is around 5 (4–8), far below CI of
cirrus cloud (being around 10 with the maximum value exceeding 20) at 250 hPa
[Vernier et al., 2015].

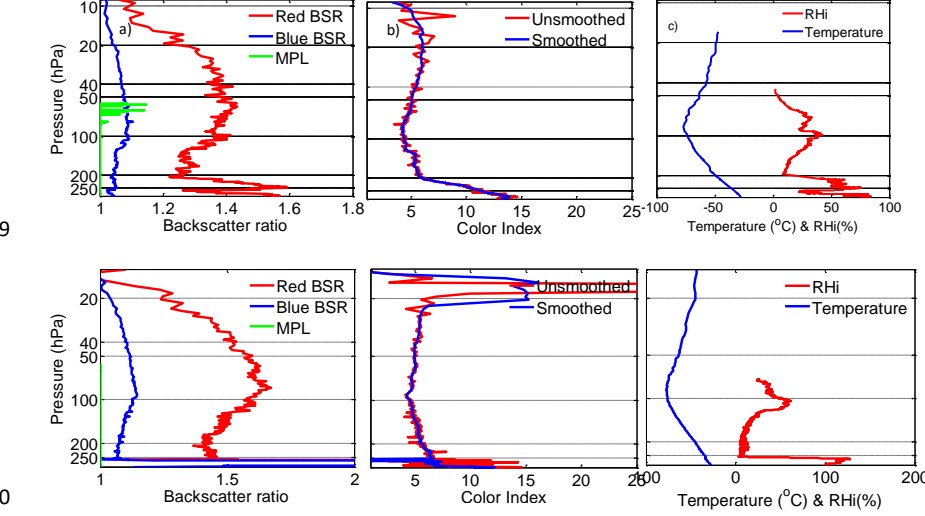




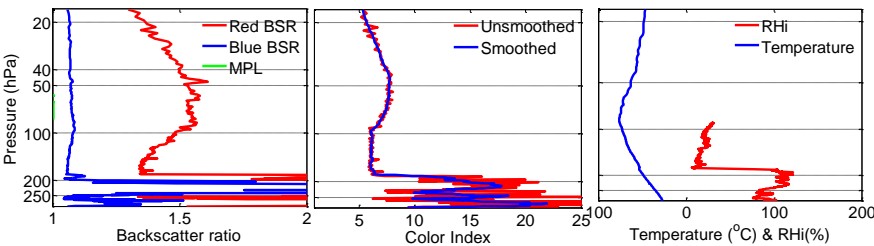

**Fig. 1.** (a) Three cases of the backscattering ratio profile from COBALD and MPL
measurements on June 18 (top), July 15 (middle) and July 25 (bottom), 2014. (b) The
calculated CI profiles from the ABSR at two wavelengths. (c) Temperature and RHi
profiles measured by the RS92 radiosonde and CFH, respectively.
Pinnick et al. [1975] adopted a lognormal distribution with a mode radius of 0.0725
µm and standard deviation ($\sigma$) of 1.86 to parameterize the background aerosols in the
stratosphere. Rosen and Kjome [1991] suggested a mode radius between 0.04 and 0.06
µm and $\sigma$ value of ~2.0-2.2 for the 20-km stratospheric aerosol background layer. In
this study, the CI as a function of mode radius was derived from Mie calculation using
a lognormal distribution for different size of aerosols with standard deviations ($\sigma$) of
1.8 and 2.2 respectively and the result is shown in Fig. 2. The CI increases
monotonously from 1 to 15 with mode radius growing from 1 nm to 1µm. The CI of
the enhanced aerosol layer from COBALD measurement usually varied from 4 to 8 as
indicated in this figure. With the assumed lognormal widths, the measured CI imposes
an upper limit of 100 nm on the particle radius. Therefore, we conclude that the
enhanced aerosol layer is composed of a large number of fine particles with radius less
than 0.1 µm. It has been documented that aerosols in the UTLS are mainly composed
of liquid inorganics with typical mode radii smaller than 0.1 µm [Tobo et al., 2007].
Our observations in Linzhi are consistent with previous findings.





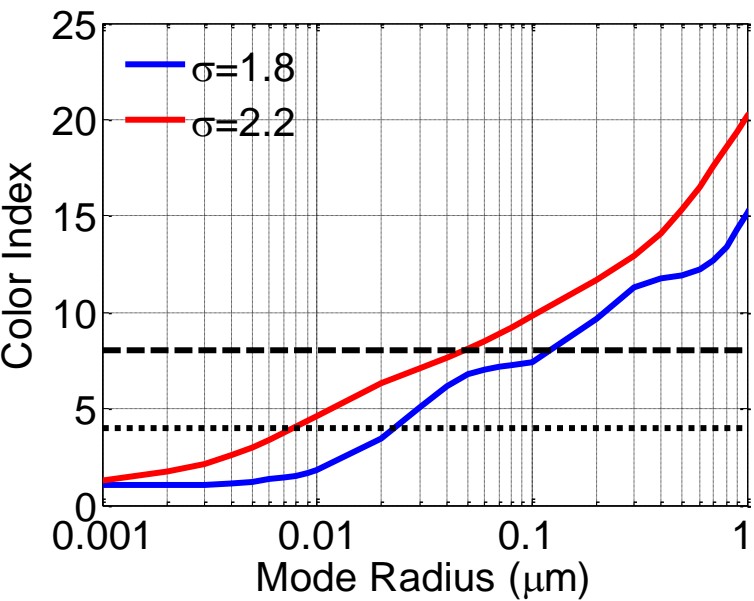

**Fig. 2**. CI as a function of mode radius from Mie calculation assuming an index of
refraction of 1.45 and a lognormal size distribution with the indicated standard
deviations (σ) of 1.8 and 2.2. The dotted and dashed lines represent the minimum (~4)
and maximum (~8) CI of the enhanced aerosol layer from COBALD measurement for
all cases.
The middle troposphere over the Tibetan Plateau is likely to act as a pipe for the
transport of water vapor from the marine boundary layer (i.e., Indian Ocean and South
China Sea) to the UTLS, leading to an increase of $H_2O$ mixing ratio near the tropopause
[Fu et al., 2006; Lelieveld et al., 2007]. Figure 3(a) presents the CFH $H_2O$ profiles from
110 hPa (~16 km ASL) to 90 hPa (~17.5 km ASL). It is noticed that $H_2O$ concentration
changes greatly in the vertical direction (3~12ppmv) for some cases. The dehydration
process results in minimum $H_2O$ concentration just above the altitude of each lowest
temperature. Pronounced decrease of the $H_2O$ concentration from 110 hPa to 90 hPa
are attributed to convection transport of moist air parcels just occurring during the
balloon flying periods. The three relatively uniform $H_2O$ profiles (on June 18, July 25
& 30) correspond to the well mixed status of strong upward transport prior to the
balloon-based measurements. The water vapor cycle driven by synoptic-scale



convection increases the possibility of the condensational growth of aerosols near the
tropopause over the Tibetan plateau. It has been estimated that the scattering ratio could
increase by 10% to 50% with a water vapor mixing ratio enhancement from 3 ppmv to
6 ppmv [Vernier et al., 2011].
Fig. 3(b) presents the variation of CI with RHi for all cases between 50 hPa and
150 hPa, the typical altitude range for the ATAL. The dependency of CI on RHi can be
classified into three types according to the CI of dry aerosols, i.e. the aerosols existing
at very low relative humidity (e.g., RHi < 20%):
(1) When the CI of dry aerosol is larger than about 6, CI of the enhanced aerosol
layer shows an exponential growth with increasing RHi;
(2) When the CI of dry aerosol is smaller than about 6, CI of the enhanced aerosol
layer decreases with increasing RHi in a slope of -0.03;
(3) When the CI of dry aerosol is close to 6, it keeps almost constant with variation
of RHi.
As the CI can be regarded as an indicator of aerosol particle size, it can be inferred
that for those aerosol particles with large dry sizes (Type 1, i.e., CI> 6), increasing RHi
facilitates water vapor and other gaseous precursors to condense onto pre-existing
aerosol particles and then contribute to the particle growth. For those with small dry
sizes (Type 2 and Type 3, i.e., CI <= 6), the situation appears to be more completed and
cannot be fully understood without more detailed information about aerosol chemical
composition and their gas precursors. Since all these aerosol particles were observed at
very low RHi, well below 40% deliquescence relative humidity of most of the salts
(e.g., 40% for $NH_4HSO4$) [Benson et al., 2009], the condensation of water vapor should
have negligible effect on the growth of these particles. New particle formation through
the gas-to-particle conversion process, which tends to become faster with increasing
RH [Fountoukis and Nenes, 2007], increases the number concentration, resulting in
decrease of mode radius of bulk aerosols. Therefore, the decrease of CI with RHi (Type
2) indicates that new particle formation might play an important role in the formation
and prevalence of fine particles in the UTLS over the Tibetan Plateau.



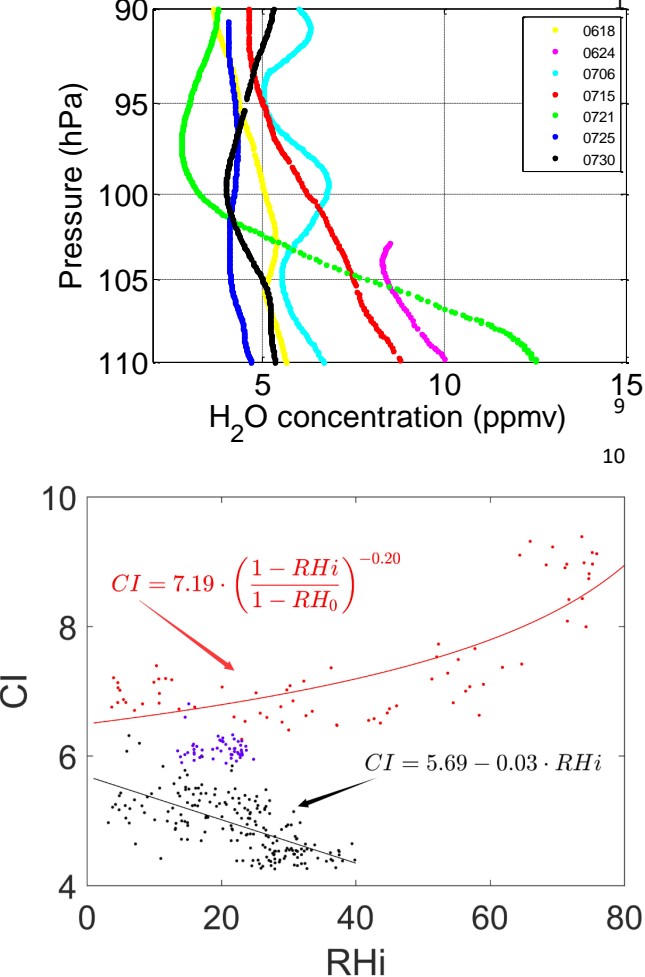

**Fig. 3**. (a) H$_2$O concentration from CFH measurements, and (b) the variation of CI with

RHi between 50 hPa and 150 hPa for all cases. The two fitted equations all exceeding

the 99% significance level.

Based on the BSR and CI at the UTLS altitudes (50-150 hPa) from COBALD, we
calculated the aerosol volume concentration in the enhanced aerosol layer for the two
typical CI variation trend according to an assumption of lognormal size distribution
with standard deviation of 1.8. The variation of aerosol volume concentration
distributions with RHi is shown in Fig. 4. It can be seen from Fig 4a that when RHi is
less than 60%, aerosol mode radius ranges mostly between 0.04 and 0.07 µm, and it



increases steeply to 0.2 µm when RHi is more than 60%. The aerosol volume
concentrations are obviously high compared with those in dry condition, especially for
those particles with a mode radius of 0.1 µm. For those aerosols with small initial dry
particle size (as shown in Fig 4b), accompanied by a mode radius decrease from 0.04
to 0.03 µm, the aerosol volume concentration increases by 4-5 times when RHi rises
from nearly zero to 40%, indicating that the number concentrations experience an
explosive growth due to the formation of new particles.

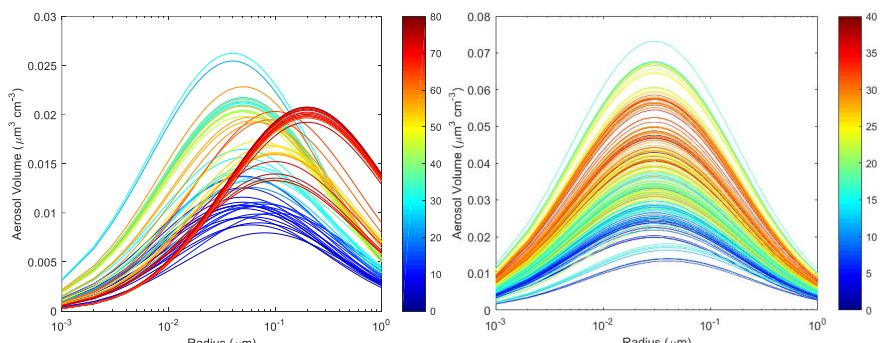

**Fig. 4**. The variation of aerosol volume concentration distributions in the enhanced
aerosol layer with RHi for (a) case 5 (July 21), and (b) the other cases corresponding to
the black dots in Fig 3b. The color of each distribution represents RHi labeled on the
color bar.
**4. Conclusions**
The vertical profiles of aerosol BSR measured over the Tibetan Plateau during
summertime demonstrate an enhanced aerosol layer, consisting predominantly of fine
particles with mode radius smaller than 0.1 µm, in the UTLS. The size of particles in
the enhanced aerosol layer shows an exponential increase with increasing RHi when
the CI of dry aerosols is larger than 6 (corresponding mode radius larger than 0.04 µm).
It can be inferred that for increasing RHi leads to the condensation of water vapor and
other gaseous precursor onto pre-existing aerosol particles and contributes to the
particle growth. For the CI of dry aerosols smaller than about 6 (i.e., mode radius
smaller than 0.04 µm), the size of particles in the enhanced aerosol layer decreases with



increasing RHi. In this case, more new particle formation, which results in a decrease of aerosol mode radius and increase of number concentration, can play an important role in the accumulation of large amounts of fine particles in the UTLS over the Tibetan Plateau. Chemical interactions involved in the stratosphere troposphere exchange are complicated and further experimental and model studies are needed to understand the nature and origin of ATAL and its influence on global atmospheric chemistry and climate.

**Author Contributions**

Qianshan He, Jianzhong Ma and Xiangdong Zheng designed the study. Holger Vömel and Frank G. Wienhold respectively contributed to data quality control of COBALD and CFH. Guangming Shi calculated Mie scattering parameters. Wei Gao, Dongwei Liu and Tiantao Cheng contributed to data analysis, numerical experiments, interpretation and paper writing. Qianshan He did further analysis and interpreted the results. All authors contributed to improve the manuscript.

*Acknowledgements.* This study was supported by the National Natural Science Foundation of China (Grant No. 91637101) and the Shanghai Science and Technology Committee Research Project (Grant No. 16ZR1431700). We thank all TOAR team members and the staff from the Tibet Meteorological Service for assisting our experiment work. We also thank Dr. Yutaka Tobo, whose useful suggestions have greatly improved the paper.

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
