# Peer review of "Observational evidence of particle hygroscopic growth in"

_Atmospheric Chemistry and Physics, 2019_

## Referee Comment (RC1) · Anonymous Referee #1 · 22 Feb 2019

Review of He et al. [2019]

This study focuses on balloon measurements carried out from the Tibetan plateau to understand the microphysical processes involved in aerosol formation and growth in the Upper Troposphere and Lower Stratosphere during the Summer Asian Monsoon. The authors use the COBALD backscatter sonde together with the Cryogenic Frost Point Hygrometer to understand how humidity affects the size of aerosols. In addition, they use Mie calculations to interpret those measurements. Overall, the paper is short, to the point, well written and follow a logical path with clear figures and consistent interpretation. I would recommend the publication of this manuscript in ACP after the following points are corrected :

- P1-L27 replace "a balloon.." by "the balloon.."

[Figure]

- P1-L28 add "COBALD sensor"

- P2L20: Park et al. [2007] should not be quoted here but rather after "large scale circulation.."

- P3L6. Frey et al. [2011] talk about the West African Monsoon and the Asian Monsoon. Do you think it's relevant here ?

- P7L7. Are you sure that the Kelud eruption did not impact those balloon measurements ?

- Fig.1. I would rather differentiate in this plot: the Junge Layer, the stratospheric aerosol layer peaking in the mid stratosphere and the ATAL which is limited to the Upper Tropospheric and Lower Stratospheric region.

- Fig.2. Here, it's important to define the lower size boundary that cannot be observed by COBALD due to the lack of scattering efficiency of small aerosols. I would say that 30-40 nm is probably the limit.

- Fig.3 was also explored in Vernier et al., 2015 (Fig.3) using the same technique. I think it's important to make sure that data plotted here are not in the stratosphere and should remain below 19 km. The upper pressure limit (50 hPa) includes stratospheric data and I believe that the points in black where CI is between 4-6 and RHi below 40 % could be in the stratosphere. It would be interesting to color the points according to their heights.

More generally, the authors should remain the reader than the conclusions drawn from this paper are only based on 3 balloon flights so that general conclusions should be established with caution.
* * *

---

## Referee Comment (RC2) · Anonymous Referee #2 · 29 Mar 2019

Review of ACP-2019-6: He et al., 'Observational evidence of particle condensational growth in the UTLS over Tibetan Plateau'

Overall, the manuscript is reasonably well written and logically constructed. The writing is certainly understandable, but does exhibit some usage and punctuation errors. The topic is appropriate for publication in ACP.

One general comment is on the use and discussion of 'condensational growth' in the context of water uptake. While perhaps technically correct, it would be better to distinguish between hygroscopic growth, a dynamic and typically reversible process, and growth of particles by accretion of additional low volatility material (e.g. $H_2SO_4$ or SOA). It is noted in the manuscript that 'the growth mechanism of the particles in the ATAL is still poorly described' and while the authors discuss observed relationships between RHi and the COBALD-derived aerosol backscatter and color index, the phenomenology does not fully constrain the mechanisms. If the elevated RHi resulted in condensational growth through enhanced chemical production, the observed relationship would break down since the dependence would be on RHi history rather than instantaneous RHi as considered here.

Specific comments:

Title:    'condensational growth' – see comment above
          'over Tibetan Plateau' >> 'over the Tibetan Plateau'

P1L24:    'Water plays an important role in the growth' – see comment above. Water is important in determining the size, and therefore radiative properties, of the particles.

P1L27:    'aerosol backscattering ratio' >> 'backscatter ratio' more common and as it does appear elsewhere in the manuscript. 'aerosol' is not appropriate here since the acronym and numbers quoted subsequently are BSR, not ABSR = BSR – 1, as defined in the paper. Alternately, 'aerosol backscattering' could be used here and 'backscatter ratio' added to BSR in L30.

P1L27:    'with a balloon-borne lightweight COBALD at Linzhi' – the profiles were measured with separate COBALD instruments, so perhaps 'using balloon-borne, lightweight COBALD instruments above Linzhi'. I believe the COBALD acronym should technically be spelled out in the abstract as well as the body (as UTLS is).

P1L32:    Here the CI is defined in terms of ABSR, but that is not defined—perhaps harmonize with the BSR discussion earlier in the abstract.

P2L1:     delete 'dominant' >> 'indicating the prevalence of fine particles'

P2L6:     as noted in the general comments above, water uptake at high RH is increasing the size of the particles (hygroscopic growth), but it is not really the case that water vapor

is playing 'a very important role in the formation of large amounts of fine particles' unless you are considering the role of H2O in aerosol nucleation, which is not something that is addressable through the measurements in this study.

P2L8:    'condensational growth' – see comment above

P2L9:    'enhancement' – hygroscopic growth would enhance the size, and therefore radiative effects, of the ATAL aerosol, but is really not responsible for the ATAL formation.

P2L22:   'global warm effect from greenhouse gases.' >> 'global warming effect from increasing greenhouse gas concentrations.'

P2L23:   'maximum' >> 'elevated' (also in L27)

P3L5:    '[Frey et al., 2011] proposed' >> 'Frey et al. [2011] proposed'

P3L9:    'after the relative humidity' – after it what? This sentence could be restructured to be clearer. Also, CALIOP is the lidar, CALIPSO is the satellite.

P3L8-12: related to the question of the use of 'condensational growth', I am unclear what is being proposed as the mechanism by which increased relative humidity would take one month to affect the size of aerosol. Beyond the role of H2O in the formation of molecular (or ion-molecule) clusters that can subsequently grow into aerosols, it would take significant supersaturations for sufficient water to condense on nanometer sized (nucleation mode) particles to produce growth, and then it would likely produce activation to large size (cloud). This could theoretically lead to growth of the underlying particles through aqueous chemistry (e.g. $SO_2 \rightarrow H_2SO_4$), resulting in larger residual particles after the humidity decreases, but that would not contribute to the backscatter – humidity relationship that is the core of the argument in the paper.

P3L14:   'mechanism' >> 'mechanisms'

P3L15:   'the coagulation…the nucleation' >> 'coagulation…nucleation'

P3L17:   'Except for coagulation,' >> 'Compared with coagulation,'

P3L21:   'the stratospheric aqueous' >> 'stratospheric aqueous'

P4L18:   it would be better to list the years of the BATAL campaign than to say 'More Recently'.

P4L25:   'the vertical profiles' >> 'vertical profiles'

P5L7:    'of the Compact Optical Backscatter AerosoL Detector (COBALD) particle backscatter sonde, the iMet and RS92 radiosonde, and the cryogenic frost-point hygrometer

(CFH).' >> 'of a Compact Optical Backscatter AerosoL Detector (COBALD) instrument, iMet and RS92 rediosondes, and a cryogenic frost-point hygrometer (CFH).'

P5L10:   'flew at' >> 'rose with' or 'ascended at a rate of'

P5L12:   'ascending' >> 'ascent'

P5L22:   delete 'follow'

P5L27:   'scatter' >> 'scattering'

P5L29:   'raw data the blue' >> 'raw data, the blue'

P6L1:    'and the precision in an order of 1%' >> 'and a precision of approximately 1%'. Rosen and Kjome should be included in a discussion of instrument uncertainty since it, to the extent that the COBALD instrument and data treatment are functionally similar, provides a far more complete description of the instrument performance than is available in the COBALD references such as Vernier et al. (2015).

P7L10:   the ATAL has been typically observed [e.g. Vernier et al. (2015)] to occupy a much narrower range of altitudes than is described here and shown Fig 1. The top here certainly extends far into the stratosphere, many km above the tropopause and into the altitude range of the Junge layer. To tie the analysis here to the ATAL it is important to discuss the nature of the layer observed during these measurements and how/why it differs so dramatically from other ATAL observations.

P9L12:   'concentration' should be 'mixing ratio' here and elsewhere (e.g. L14, L15, Fig 3 caption)

P9L15:   'convection transport' >> 'convective transport'

P10L6:   'dependency' >> 'dependence'

---

## Author Comment (AC1) · 10 May 2019

**Reviewer #1:**

This study focuses on balloon measurements carried out from the Tibetan plateau to understand the microphysical processes involved in aerosol formation and growth in the Upper Troposphere and Lower Stratosphere during the Summer Asian Monsoon. The authors use the COBALD backscatter sonde together with the Cryogenic Frost Point Hygrometer to understand how humidity affects the size of aerosols. In addition, they use Mie calculations to interpret those measurements. Overall, the paper is short, to the point, well written and follow a logical path with clear figures and consistent interpretation. I would recommend the publication of this manuscript in ACP after the following points are corrected :

- P1-L27 replace "a balloon.." by "the balloon.."
**R: Done.**

- P1-L28 add "COBALD sensor"
**R: We rewrote the sentence as '***using the balloon-borne, lightweight Compact Optical Backscatter AerosoL Detector (COBALD) instruments above Linzhi***' according to the comments from the two reviewers..**

- P2L20: Park et al. [2007] should not be quoted here but rather after "large scale circulation.."
**R: Done.**

- P3L6. Frey et al. [2011] talk about the West African Monsoon and the Asian Monsoon.
Do you think it's relevant here ?
**R: We agreed with reviewer's concerns and rewrote this and related sentence earlier as '***Sources and formation mechanism of aerosols in the UTLS, especially over the tropics, have been studied over the past decades. New particle formation events can occur at very low temperatures accompanied by the outflow of convective systems, as observed in the West African Monsoon [Frey et al., 2011]***'. Here we merely give a brief overview about the studies for the UTLS.**

- P7L7. Are you sure that the Kelud eruption did not impact those balloon measurements?
**R: We appreciate the reviewer's suggestions (also from Reviewer #2) and add the corresponding discussion about the Kelud eruption.**
*'On February 13, 2014 the Mt. Kelud (8°S, 112°E) in Indonesia erupted, with a volcanic plume located near 18-21 km within the tropical stratosphere, which was detected 11 days after the eruption by the Cloud-Aerosol Lidar with Orthogonal Polarization (CALIOP) onboard the Cloud-Aerosol Lidar and Infrared Pathfinder Satellite Observation (CALIPSO) [Vernier et al, 2016]. Stratospheric aerosols were perturbed significantly by the Kelud volcanic plumes, especially the fresh ash plume in the southern hemisphere [Vernier et al, 2016; Sakai et al., 2016], The Kelud volcanic eruption might have negligible influence on the observed aerosols in the ATAL, since the ATAL began to form about four months after the Kelud eruption when the volcanic materials in the troposphere might have vanished. On the other hand, CALIOP data analysis also showed that sulfate components from the Kelud volcanic eruption, peaking at an higher altitude with a longer residence time compared with the volcanic ashes, influenced aerosol optical depth (AOD) between 20°N and 20°S 18-25 km considerably three months after the eruption [Vernier et al, 2016]. It is*

*likely that sulfate aerosols from the Kelud eruption contributed to stratospheric background aerosols above the ATAL and even in the Junge layer at slightly higher latitude, as indicated by our COBALD measurements'.*

- Fig.1. I would rather differentiate in this plot: the Junge Layer, the stratospheric aerosol layer peaking in the mid stratosphere and the ATAL which is limited to the Upper Tropospheric and Lower Stratospheric region.

**R: We appreciate the reviewer's suggestions and add the corresponding statement about the enhanced aerosol layer from COBALD measurement in the first paragraph of Sect. 3.**

*"The enhanced aerosol layer from COBALD measurement is a mixture of ATAL and the on-setting Junge Layer due to the signal above 50 hPa stemming from the Junge Layer but the maximum occurring in ATAL".*

- Fig.2. Here, it's important to define the lower size boundary that cannot be observed by COBALD due to the lack of scattering efficiency of small aerosols. I would say that 30-40 nm is probably the limit.

**R: We agreed with reviewer's concerns and added the corresponding statement and a new table about the lower size boundary of COBALD measurement.**

*"The signal to noise ratio at the blue channel with respect to the molecular Rayleigh backscatter at tropopause conditions (taken 100 hPa and 210 K) is 220. Given the molecular backscatter coefficient of $4.4e^{-7}$ ($sr^{-1}m^{-1}$) for 455 nm, this corresponds to a backscatter coefficient minimum detection limit of $2e^{-9}$ ($sr^{-1}m^{-1}$), which is holding in general over the entire profile. To define an aerosol size limit, typical aerosol dumber densities need to be assumed: 10 $cm^{-3}$ for stratospheric background and 100 $cm^{-3}$ for the ATAL. The aerosol backscatter coefficients of different aerosol mode radius for the typical aerosol dumber densities are calculated by Mie theory and listed in Table 1. The results confirm that the particles with 100 nm radius are well detected under background conditions, which mainly contribute to the particulate backscatter ratio of approx. 0.01 and is always present. With increasing particle number density, the particles with 30 nm radius start to contribute to the particulate backscatter ratio ($> 2e^{-9}$ $sr^{-1}m^{-1}$). Therefore, the lower size boundary that cannot be observed by COBALD due to the lack of scattering efficiency of small aerosols can be defined as 30 nm.*

**Table 1** The aerosol backscatter coefficients of different aerosol mode radius for the typical aerosol dumber densities.

| Mode Radius (nm) | 10 | 30 | 100 |
|---|---|---|---|
| $\beta_a$@10 $cm^{-3}$ ($sr^{-1}$ $m^{-1}$) | $1e^{-12}$ | $3e^{-10}$ | $2e^{-8}$ |
| $\beta_a$@100 $cm^{-3}$ ($sr^{-1}$ $m^{-1}$) | $1e^{-11}$ | $3e^{-9}$ | $2e^{-7}$ |

*"*

- Fig.3 was also explored in Vernier et al., 2015 (Fig.3) using the same technique. I think it's important to make sure that data plotted here are not in the stratosphere and

should remain below 19 km. The upper pressure limit (50 hPa) includes stratospheric data and I believe that the points in black where CI is between 4-6 and RHi below 40 % could be in the stratosphere. It would be interesting to color the points according to their heights.

**R: We agreed with reviewer's concerns and replotted Fig 3b by coloring the points according to the height of particles.**

More generally, the authors should remain the reader than the conclusions drawn from this paper are only based on 3 balloon flights so that general conclusions should be established with caution.

**R: The reviewer's comments are very valuable. We added a sentence about the conclusions drawn from only 7 balloon flights.**

*"It must be borne in mind that the conclusions drawn from this study are only based on 7 balloon flights so that general conclusions should be established with caution".*

---

## Author Comment (AC2) · 10 May 2019

Reviewer #2:

Overall, the manuscript is reasonably well written and logically constructed. The writing is certainly understandable, but does exhibit some usage and punctuation errors. The topic is appropriate for publication in ACP.

One general comment is on the use and discussion of 'condensational growth' in the context of water uptake. While perhaps technically correct, it would be better to distinguish between hygroscopic growth, a dynamic and typically reversible process, and growth of particles by accretion of additional low volatility material (e.g. H2SO4 or SOA). It is noted in the manuscript that 'the growth mechanism of the particles in the ATAL is still poorly described' and while the authors discuss observed relationships between RHi and the COBALD-derived aerosol backscatter and color index, the phenomenology does not fully constrain the mechanisms. If the elevated RHi resulted in condensational growth through enhanced chemical production, the observed relationship would break down since the dependence would be on RHi history rather than instantaneous RHi as considered here.

**R: We thank Reviewer #2 for insightful comments and constructive suggestions! We agree with reviewer's concerns and have revised the manuscript accordingly as specified below.**

Specific comments:

Title: 'condensational growth' – see comment above

'over Tibetan Plateau' >> 'over the Tibetan Plateau'

**R: The title of this manuscript has been changed to '***Observational evidence of particle hygroscopic growth in the UTLS over the Tibetan Plateau***'.**

P1L24: 'Water plays an important role in the growth' – see comment above. Water is important in determining the size, and therefore radiative properties, of the particles.

**R: We rewrote this sentence as suggested by the reviewer.**

P1L27: 'aerosol backscattering ratio' >> 'backscatter ratio' more common and as it does appear elsewhere in the manuscript. 'aerosol' is not appropriate here since the acronym and numbers quoted subsequently are BSR, not ABSR = BSR – 1, as defined in the paper. Alternately, 'aerosol backscattering' could be used here and 'backscatter ratio' added to BSR in L30.

**R: We changed '***aerosol backscattering ratio***' to '***backscatter ratio***' as suggested by the reviewer.**

P1L27: 'with a balloon-borne lightweight COBALD at Linzhi' – the profiles were measured with separate COBALD instruments, so perhaps 'using balloon-borne, lightweight COBALD instruments above Linzhi'. I believe the COBALD acronym should technically be spelled out in the abstract as well as the body (as UTLS is).

**R: We rewrote this sentence as '***using the balloon-borne, lightweight Compact Optical Backscatter AerosoL Detector (COBALD) instruments above Linzhi***' according to the comments from both reviewers.**

P1L32: Here the CI is defined in terms of ABSR, but that is not defined—perhaps harmonize with

the BSR discussion earlier in the abstract.

**R: We rewrote this sentence according to this comment as well that for P1L27 about BSR.**

P2L1: delete 'dominant' >> 'indicating the prevalence of fine particles'
**R: Done.**

P2L6: as noted in the general comments above, water uptake at high RH is increasing the size of the particles (hygroscopic growth), but it is not really the case that water vapor is playing 'a very important role in the formation of large amounts of fine particles' unless you are considering the role of H2O in aerosol nucleation, which is not something that is addressable through the measurements in this study.

**R: We rewrote this phrase as** '*water vapor can play a very important role in increasing the size of fine particles*'**.**

P2L8: 'condensational growth' – see comment above
**R: We replaced '***condensational growth***' with '***hygroscopic growth***'.**

P2L9: 'enhancement' – hygroscopic growth would enhance the size, and therefore radiative effects, of the ATAL aerosol, but is really not responsible for the ATAL formation.

**R: We rewrote this sentence as '***aerosol particle hygroscopic growth is an important factor influencing the radiative properties of the Asian Tropopause Aerosol Layer (ATAL) during the Asian summer monsoon***'.**

P2L22: 'global warm effect from greenhouse gases.' >> 'global warming effect from increasing greenhouse gas concentrations.'
**R: Done.**

P2L23: 'maximum' >> 'elevated' (also in L27)
**R: Done.**

P3L5: '[Frey et al., 2011] proposed' >> 'Frey et al. [2011] proposed'
**R: This sentences has been changed as '***New particle formation events can occur at very low temperatures accompanied by the outflow of convective systems, as observed in the West African Monsoon [Frey et al., 2011]***' according to the comment from Reviewer #1.**

P3L9: 'after the relative humidity' – after it what? This sentence could be restructured to be clearer. Also, CALIOP is the lidar, CALIPSO is the satellite.
**R: We meant** *a one-month phase lag of the starting point of increase in aerosol scattering ratio after the starting point of increase in relative humidity*. **Anyhow, we deleted this sentence considering that it dose not contribute to the core of the argument in this paper, as pointed out by the reviewer.**

P3L8-12: related to the question of the use of 'condensational growth', I am unclear what is being proposed as the mechanism by which increased relative humidity would take one month

to affect the size of aerosol. Beyond the role of H2O in the formation of molecular (or ion-molecule) clusters that can subsequently grow into aerosols, it would take significant supersaturations for sufficient water to condense on nanometer sized (nucleation mode) particles to produce growth, and then it would likely produce activation to large size (cloud). This could theoretically lead to growth of the underlying particles through aqueous chemistry (e.g. SO2 & H2SO4), resulting in larger residual particles after the humidity decreases, but that would not contribute to the backscatter – humidity relationship that is the core of the argument in the paper.

**R: We agreed with reviewer's concerns and deleted this sentence accordingly. To make the issue more clearly, we added the following sentences to the last second paragraph of Sect. 1:** **'***New particle formation and growth of particles by accretion of additional low volatility materials (e.g., $H_2SO_4$) tend to be an irreversible but slow progress due to limited amount of condensable gases, In contrast, hygroscopic growth of particles is a dynamic and typically reversible process, and may affect the size of particles and its variation in the ATAL more remarkably in a relatively short time since sufficient amount of water vapor can be frequently lofted to the UTLS via deep convection during the Asian monsoon [Fu et al., 2006]***'.**

P3L14: 'mechanism' >> 'mechanisms'
**R: Done.**

P3L15: 'the coagulation…the nucleation' >> 'coagulation…nucleation'
**R: Done.**

P3L17: 'Except for coagulation,' >> 'Compared with coagulation,'
**R: Done.**

P3L21: 'the stratospheric aqueous' >> 'stratospheric aqueous'
**R: Done.**

P4L18: it would be better to list the years of the BATAL campaign than to say 'More Recently'.
**R: We rewrote this sentence as '***A series of balloon borne activities between 2014 and 2017***' as suggested by the reviewer.**

P4L25: 'the vertical profiles' >> 'vertical profiles'
**R: Done.**

P5L7: 'of the Compact Optical Backscatter AerosoL Detector (COBALD) particle backscatter sonde, the iMet and RS92 radiosonde, and the cryogenic frost-point hygrometer (CFH).' >> 'of a Compact Optical Backscatter AerosoL Detector (COBALD) instrument, iMet and RS92 rediosondes, and a cryogenic frost-point hygrometer (CFH).'
**R: Done.**

P5L10: 'flew at' >> 'rose with' or 'ascended at a rate of'
**R: Done.**

P5L12: 'ascending' >> 'ascent'
**R: Done.**

P5L22: delete 'follow'
**R: Done.**

P5L27: 'scatter' >> 'scattering'
**R: Done.**

P5L29: 'raw data the blue' >> 'raw data, the blue'
**R: This sentence was deleted owing to our response to the reviewer's comment for P6L1.**

P6L1: 'and the precision in an order of 1%' >> 'and a precision of approximately 1%'. Rosen and Kjome should be included in a discussion of instrument uncertainty since it, to the extent that the COBALD instrument and data treatment are functionally similar, provides a far more complete description of the instrument performance than is available in the COBALD references such as Vernier et al. (2015).
**R: The reviewer's comments are very valuable. We added a description of the instrument performance to the first paragraph of Sect. 2.1 as '*Before flight, the signal from each backscatter sonde is compared with a dedicated set of four standard backscatter sondes maintained in Laramie. The repeatability of the relative calibration between backscatter sondes is about $\pm 1\%$. The absolute calibration is believed accurate to better than $\pm 3\%$. Since naturally occurring aerosol backscatter ratios may be quite low, especially in the blue channel, it is important to consider potential sources of error and uncertainty in the absolute values derived from the basic measurements themselves. In the blue channel, a conservative adjustment procedure has been made in the range of 0 to 4% to eliminate nonphysical average values occurring in the troposphere [Rosen et al., 1997]'.***

P7L10: the ATAL has been typically observed [e.g. Vernier et al. (2015)] to occupy a much narrower range of altitudes than is described here and shown Fig 1. The top here certainly extends far into the stratosphere, many km above the tropopause and into the altitude range of the Junge layer. To tie the analysis here to the ATAL it is important to discuss the nature of the layer observed during these measurements and how/why it differs so dramatically from other ATAL observations.
**R: We appreciate the reviewer's suggestions (also from Reviewer #1) and add the corresponding statement about the enhanced aerosol layer from COBALD measurement.**
*'On February 13, 2014 the Mt. Kelud (8°S, 112°E) in Indonesia erupted, with a volcanic plume located near 18-21 km within the tropical stratosphere, which was detected 11 days after the eruption by the Cloud-Aerosol Lidar with Orthogonal Polarization (CALIOP) onboard the Cloud-Aerosol Lidar and Infrared Pathfinder Satellite Observation (CALIPSO) [Vernier et al, 2016]. Stratospheric aerosols were perturbed significantly by the Kelud volcanic plumes, especially the fresh ash plume in the southern hemisphere [Vernier et al, 2016; Sakai et al., 2016], The Kelud volcanic eruption might have negligible influence on the observed aerosols in the ATAL, since the*

*ATAL began to form about four months after the Kelud eruption when the volcanic materials in the troposphere might have vanished. On the other hand, CALIOP data analysis also showed that sulfate components from the Kelud volcanic eruption, peaking at an higher altitude with a longer residence time compared with the volcanic ashes, influenced aerosol optical depth (AOD) between 20°N and 20°S 18-25 km considerably three months after the eruption [Vernier et al, 2016]. It is likely that sulfate aerosols from the Kelud eruption contributed to stratospheric background aerosols above the ATAL and even in the Junge layer at slightly higher latitude, as indicated by our COBALD measurements'.*

P9L12: 'concentration' should be 'mixing ratio' here and elsewhere (e.g. L14, L15, Fig 3 caption)
**R: Done.**

P9L15: 'convection transport' >> 'convective transport'
**R: Done.**

P10L6: 'dependency' >> 'dependence'
**R: Done.**